# Correlation Analysis between Muskrat (*Ondatra zibethicus*) Musk and Traditional Musk

**DOI:** 10.3390/ani13101678

**Published:** 2023-05-18

**Authors:** Xin Shi, Dejun Zeng, Guijun Zhao, Chenglu Zhang, Xiaolan Feng, Chengli Zheng, Diyan Li, Ming Zhang, Hang Jie

**Affiliations:** 1Institute of Animal Genetics and Breeding, College of Animal Science and Technology, Sichuan Agricultural University, Chengdu 611130, China; 2Bio-Resource Research and Utilization Joint Key Laboratory of Sichuan and Chongqing, Chongqing Institute of Medicinal Plant Cultivation, Nanchuan, Chongqing 408435, China; 3Sichuan Institute of Musk Deer Breeding, Chengdu 611845, China; 4School of Pharmacy, Chengdu University, Chengdu 610106, China

**Keywords:** muskrat, forestry musk deer, musk, metabolites, age

## Abstract

**Simple Summary:**

Traditional musk is a scarce resource of traditional Chinese medicine in China, which is difficult to obtain. Few studies have been reported on the search for alternatives to musk. To this end, we collected muskrat musk, which is widely regarded as a substitute for musk, and performed Gas Chromatography Time-of-Flight Mass Spectrometry (GC–TOF–MS) metabolomics analysis together with musk. The results showed that muskrat musk and musk had certain similarities in terms of their composition, but the important active components of muscone were missing. Analysis of muskrat musk at different ages showed that amino acids, fatty acids, and steroids were significantly enriched, which may affect the function of muskrat scent glands through local enrichment for a short time.

**Abstract:**

Muskrat musk is considered to be a potential substitute for traditional musk. However, little is known about the similarity between muskrat musk and musk, and whether it is related to muskrat age. In this study, muskrat musk (MR1, MR2, and MR3) were from 1, 2, and 3-year-old muskrats, respectively, and white musk (WM) and brown musk (BM) were picked from male forest musk deer. The results indicated that muskrat musk had higher similarity to WM than BM. Further research showed that RM3 had the highest matched degree with WM. By significantly different metabolite analysis, we found that 52 metabolites continue to increase from 1- to 3-year-old muskrats. In total, 7 and 15 metabolites were significantly decreased in RM1 vs. RM2 and RM2 vs. RM3, respectively. Meanwhile, 30 and 17 signaling pathways were observed from increased and decreased metabolites, respectively. The increased metabolites mainly entailed enrichment in amino acid biosynthesis and metabolism, steroid hormone biosynthesis, and fatty acid biosynthesis. In conclusion, muskrat musk from three-year-old muskrat is a relatively good substitute for white musk, and the result also implies that these biological processes of amino acid biosynthesis and metabolism, steroid hormone biosynthesis, and fatty acid biosynthesis are beneficial to the secretion of muskrat musk.

## 1. Introduction

Forest musk deer, a kind of small ruminant, is famous for its musk secreted in the musk pot of adult male individuals. Musk has a variety of pharmacological activities, such as playing a role in cerebrovascular diseases and cardiovascular diseases as well as in the nervous system and cancer [1,2,3,4,5]. In general, mature musk is a kind of dark brown solid in the same way as earwax [6], whereas the immature musk in the musk pot was white and stench and it is still different in composition from the mature musk [7,8]. However, because of habitat destruction and overhunting for its marvelous effect on musk, the wild population declined sharply. Until now, forest musk deer has been listed as an endangered species in the International Union for the Conservation of Nature (IUCN) Red List [9] and Critically Endangered (CR) in the Red List of China’s Vertebrates [10]. In order to meet the needs of musk application in traditional Chinese medicine based on the protection of forest musk deer, exploring alternatives to musk is particularly necessary at present.

The muskrat (*Ondatra zibethicus* L.) is the only species in the genus Ondatra (family *Cricetidae*) which is native to Canada and North America and has been introduced to Europe, Asia, and South America [11]. Muskrats were introduced to China in the 1950s [12]. As an alien species, it is not on the endangered animal protection list. At present, muskrats are mainly distributed in Xinjiang, Heilongjiang, Shaanxi, and other provinces in China and have not formed a certain industrial scale. Its most striking feature is the existence of the male muskrat tail, abdomen, and skin between a pair of secreting glands. Muskrat is typically a seasonal breeder and its annual life cycle includes the breeding season (sexually active period, March to October) and the nonbreeding season (November to February) [13]. During the breeding season, male muskrats are willing to mate and thus produce mature sperm. Moreover, the secreting glands become greatly enlarged to secrete muskrat musk to attract females. Chemical analysis has indicated that the muskrat musk contains a variety of ingredients, such as muscone, normuscone, macrocyclic ketones, fatty acids, esters, and alcohols [14,15]. Muskrat musk can be used to treat pain and inflammatory responses caused by stroke, swelling, and abscesses with similar efficacy as musk and may be an ideal substitute for musk [16].

Compared with forest musk deer, muskrats have low environmental requirements and can reproduce quickly. More importantly, the main components of muskrat musk are similar to those of natural musk [17,18,19]. Therefore, to find the best alternatives to musk, it is imperative to understand the differences in composition between muskrat musk and different types of musk. The purpose of this research was to investigate the metabolomic change trend of muskrat musk at different ages by GC–TOF–MS technology, as well as to compare it with different types of musk. These results will provide a reference to determine the best time to collect musk from muskrats and may also offer insights into the change in feed formula during different periods.

## 2. Materials and Methods

### 2.1. Animals

Ten 1-year-old male muskrats were fed in separate 70 cm high, 130 cm long, and 60 cm wide enclosures in the Hubei Yingshan Muskrat Breeding Base (Hubei Province, China). The space layout of each enclosure included a swimming pool, a feeding platform, and a lounge. The same concentrated feed and green feed were given to musk deer and muskrats. The concentrated feed consisted of 60% corn, 20% soybean meal, 2.0% fish meal, 6% wheat bran, 9% mulberry leaf grass meal, 0.4% salt, 1.5% CaCO3, 0.8% CaHPO4, 0.15% vitamin, and 0.15% mineral. Green forages was administered according to the seasons: chicory leaves, sweet potato vines, etc., in summer and pumpkin, radish, etc., for winter. All animals were treated according to the National Animal Welfare Legislation. Experimental procedures were approved by the Institutional Animal Care and Use Committee of Sichuan Agricultural University (permit number: S20151006). Only six male muskrats survived during the entire experimental period.

### 2.2. Sampling

A total of 18 samples (1.2 g each) from 6 muskrat individuals were collected in mid-June for 3 consecutive years (2017–2019), at which time the secretion activity reached its peak. The muskrat musk samples were divided into three groups according to age. Each group contained six biological replicates: RM1, RM2, and RM3 were from 1, 2, and 3-year-old muskrats, respectively. WM and BM were used as controls from previous collections in the forest musk deer breeding base (Chongqing, China). Samples were placed in sterile cryogenic vials, stored in liquid nitrogen, and transported to Sichuan Agricultural University where they remained frozen at −80 °C until metabolite identification.

### 2.3. Sample Preparation

Each 50 ± 1 mg (BSA124S-CW, Sartorius, Göttingen, Germany) sample was transferred into a 2 mL tube and 500 μL of precooled extraction mixture (methanol/chloroform (*v*:*v*) = 3:1) containing 10 μL of internal standard (adonitol, 0.5 mg/mL stock solution) was added, following by vortexing for 30 s. Steel balls were added for processing with a 35 Hz grinding machine (JXFSTPRP-24, Shanghai Jingxin Technology Co., Ltd., Shanghai, China) for 4 min and then ultrasonicated (YM-080S, Shenzhen Fangao Microelectronics Co., Ltd., Shenzhen, China) with an ice water bath (Forma 900 series, Thermo Fisher Scientific, Waltham, MA, USA) for 5 min (this procedure was repeated twice). After centrifugation (Heraeus Fresco17, Thermo Fisher Scientific) at 4 °C for 15 min at 12,000 rpm, 200 μL of the supernatant was transferred to a fresh tube. To prepare the QC (quality control) sample, 40 μL of each sample was removed and combined. After evaporation in a vacuum concentrator (LNG-T98, Taicang Huamei Biochemical Instrument Factory, Taicang, China), 60 μL of methoxyamination hydrochloride (20 mg/mL in pyridine) was added and followed by incubation at 80 °C (DHG-9023A, Shanghai Yiheng Scientific Instrument Co., Ltd., Shanghai, China) for 30 min followed by derivatization with 80 μL of BSTFA reagent (1% TMCS, *v*/*v*) at 70 °C for 1.5 h. Samples were gradually cooled to room temperature and 8 μL of FAMEs (in chloroform) was added to the QC sample.

### 2.4. GC–TOF–MS Analysis

GC–TOF–MS analysis was performed using an Agilent 7890 gas chromatograph (7890A, Agilent) coupled with a time-of-flight mass spectrometer (PEGASUS HT, LECO, Beijing, China). The system utilized a DB-5MS capillary column (DB-5MS (30 m × 250 μm × 0.25 μm), Agilent, Santa Clara, CA, USA). A 1 μL aliquot of each sample was injected in splitless mode. Helium was used as the carrier gas, the front inlet purge flow was 3 mL/min, and the gas flow rate through the column was 1 mL/min. The initial temperature was maintained at 50 °C for 1 min, raised to 310 °C at a rate of 10 °C/min, and then maintained for 8 min at 310 °C. The injection, transfer line, and ion source temperatures were 280, 280, and 250 °C, respectively. The energy was −70 eV in the electron impact mode. Mass spectrometry data were acquired in the full scan mode with an *m*/*z* range of 50–500 at a rate of 12.5 spectra per second after a solvent delay of 6.35 min.

### 2.5. Data Processing

Raw data analysis, including peak extraction, baseline adjustment, deconvolution, alignment, and integration, was completed with Chroma TOF software (V 4.3x, LECO, Saint Joseph, MI, USA), and the LECO-Fiehn Rtx5 database was used for metabolite identification by matching the mass spectrum with the retention index [20]. Finally, the peaks detected in less than half of the QC samples or with a Relative Standard Deviation (RSD) >30% in the QC samples were removed. The resulting three-dimensional data, including peak number, sample name, and normalized data were imported into SIMCA software (V16.0.2, Sartorius Stedim Data Analytics AB, Umea, Sweden). Principal component analysis (PCA) and orthogonal projections to latent structures–discriminant analysis (OPLS–DA) models were tested for all samples. In general, the closer the values of R^2^Y and Q^2^ are to 1, the better the model should be. The OPLS–DA model was employed with the first principal component of VIP (variable importance in the projection) values (VIP > 1) combined with Student’s *t*-test (*t*-test) (*p* < 0.05) to find significantly different metabolites (SDMs). In addition, commercial databases, including KEGG (http://www.genome.jp/kegg/, accessed on 15 November 2022) and MetaboAnalyst 5.0 (http://www.metaboanalyst.ca/, accessed on 10 November 2022) were used for pathway enrichment analysis.

## 3. Results

### 3.1. Identification and Quantification of Compounds by GC–TOF–MS

To make our compound test more reliable, we used GC–TOF–MS to examine the quality control samples before analyzing the formal samples. The total ion chromatographic peak retention times and peak areas of all QC samples overlapped well, which indicated that the analytical system was stable (Appendix A). In addition, there were no significant peaks detected in the blank samples, indicating that the substance residue control was good and there was no cross-contamination between samples (Appendix A). Using the interquartile range denoising method, missing values in the raw dataset were filled by half of the minimum value and the internal standard normalization method was applied. A total of 632 valid peaks were identified based on GC–TOF–MS. These effective peaks were then mapped to the KEGG COMPOUND metabolomics library for identification. After removing duplicates and unknown substances, 272 metabolites expressed in 5 samples were obtained (Appendix A). The results showed that amino acids and their derivatives accounted for 20.59% of the metabolites, followed by fatty acids, carbohydrates, and steroids which were 19.49%, 13.60%, and 8.82%, respectively (Appendix A).

### 3.2. Multivariate Analysis of the Metabolomics Data

The data generated by metabolomics analysis are highly dimensional and of high complexity. Appropriate tools, such as principal component analysis (PCA) and orthogonal PLS (OPLS), need to be employed to summarize and visualize the data. Here, we used SIMCA software (V16.0.2, Sartorius Stedim Data Analytics AB, Umea, Sweden) to build a PCA model for sorting the data, and it was found that RM1, RM2, RM3, WM, and BM group clusters were clearly clusters in each PCA fractional scatter diagram (Figure 1a,d,g,j). In addition, we further used the OPLS–DA statistical method to obtain more reliable intergroup differences between the metabolites. As shown in Figure 1b,e,h,k, OPLS–DA showed a clear separation between the different groups. We also carried out a permutation test of the OPLS–DA model by randomly changing the classification variable Y (*n* = 200). The results were as follows: R^2^Y = 0.981, Q^2^ = −0.941 (RM1 vs. RM2); R^2^Y = 0.981, Q^2^ = −0.958 (RM1 vs. RM3); R^2^Y = 0.943, Q^2^ = −0.878 (RM2 vs. RM3); and R^2^Y = 0.998, Q^2^ = −0.948 (WM vs. BM) (Figure 1c,f,i,l). Thus, the above results show that the original model is robust, does not show overfitting, and can be used in the subsequent analysis.

### 3.3. Correlation Analysis between Muskrat Musk and Musk

In order to explore the relationship between muskrat musk of different maturity and musk, we used the relative content of metabolites identified to do correlation analysis. Interestingly, we found that muskrat musk at all three time points showed a high correlation with WM and a low correlation with BM. When opposed to RM, BM had a greater correlation with WM. (Figure 2a). The results of cluster analysis also showed that the composition of WM was similar to that of muskrat musk of different ages (Figure 2a). At the same time, we note that the component content between RM2 and RM3 is much closer than that of RM1 (Figure 2a).

Then, we selected the top 20 metabolites in each sample according to the relative content ranking for comparison (Table 1). They accounted for 77.67%, 80.49%, 80.12%, 83.24%, and 77.21% of the composition, respectively. We found that alanine, stearic acid, 4-oxoproline, oleic acid, glycine, and palmitic acid are common to muskrat musk and musk. They are amino acids and fatty acid compounds. There are six kinds of special components in WM and seven kinds in BM (Figure 2b). It suggests a difference in composition between musk and muskrat musk, as well as between WM and BM. This is consistent with the above correlation analysis and cluster analysis results. To find the muskrat musk that better matches for musk, we analyzed the relative content of the six metabolites between groups. Palmitic acid, stearic acid, alanine, 4-Oxoproline, and glycine increased significantly in muskrats with increasing age (*p* < 0.05) (Figure 3a,c,d,e,f). According to the relative content statistics of the main components, the content of muskrat musk in three time periods is more similar to that of WM (Figure 3a–f). Next, we took the top 20 metabolites of musk as controls to analyze the correlation between muskrat musk and musk. The results showed that WM had a higher correlation with muskrat musk and RM3 had the highest correlation with WM (Figure 4).

### 3.4. Metabolic Profiles of the Muskrats at Different Ages

Here, we used untargeted metabolomics based on GC–TOF–MS to obtain the metabolite profiles of muskrats at different ages. Specifically, according to the standard *p* < 0.05 and VIP > 1, 131 metabolites were identified and classified as SDMs among the muskrats of different ages. Between the RM1 and RM2 groups, a total of 96 SDMs were identified, of which 89 SDMs were increased and 7 SDMs were decreased (Figure 5a). A total of 65 SDMs were increased and 15 SDMs were decreased between the RM2 and RM3 groups (Figure 5b). In addition, cluster analysis also found that metabolites showed different expression patterns in different groups (Figure 5c,d). Thus, we concluded that the metabolites of muskrat musk at different ages are significantly different and may affect the medicinal value of muskrats.

To determine the effect of age on muskrat musk metabolism pathways, we used the MetaboAnalyst 5.0 database for KEGG pathway analysis. A total of 52 metabolites increased with age. They are concentrated in 30 metabolic pathways, among which the main metabolic pathways are amino acid biosynthesis and metabolism, steroid hormone biosynthesis, and fatty acid biosynthesis (Figure 5e). Another 22 metabolites declined with age. Arginine biosynthesis and fructose and mannose metabolism are the most important of the 17 metabolic pathways (Figure 5f).

## 4. Discussion

Here, for the first time, we used metabolomics to study the changing trend of muskrat musk composition at different ages and compared it with WM and BM, revealing the difference between muskrat musk and musk composition.

We obtained 632 effective peaks by GC–TOF–MS analysis, which were mapped to the KEGG database (after excluding mismatched compounds) and 272 accurate substances were identified. Muscone is considered to be the main active component in musk secreted by male musk deer [21]. Studies have shown that it has anti-inflammatory [22,23] and analgesic [24] effects. In fact, muskrat musk, as a substitute for musk, has been the focus of attention whether it contains muscone or not and has been controversial [17,25,26]. However, muscone was not found among the co-expressed metabolites we identified but a large number of fatty acid compounds (19.49%) were present. Van Dorp et al. suggested that fatty acids may be precursors of macrocyclic ketones [26]. Moreover, research has revealed that (Z)-g-cycloheptadecen-1-one (Hp) (17:1, muskrats) has greater anti-tumor activity than 3-methylcyclopentadecanone (16:0, musk deer), suggesting that even though the two species have different active ingredients, they may still have the same pharmacological effect [5]. Future studies should explore the bioactive macrocyclic ketones of muskrat musk and the mechanism of their synthesis. More importantly, we should investigate the muskrat’s defense mechanism. In order to preserve musk deer, muskrats should not be exterminated; instead, a breeding system that prioritizes animal welfare should be implemented.

Subsequently, we analyzed the composition differences between muskrat musk of different ages and musk of different types. The whole sample correlation analysis and cluster analysis showed a higher matched degree between muskrat musk and WM. Our previous studies confirmed the existence of microbial differences in the musk pods of mated and unmated forest musk deer that may be involved in musk odor fermentation. Li et al. collected musk at different secretory periods for microbiological analysis and found that microbial richness showed a decreasing trend from the early secretory stage to the late secretory stage, which was significantly related to the maturity of musk (*p* < 0.05) [6]. The study by Wang et al. found that the microbial phyla differed between WM and BM, indicating that the microbiota may influence the maturation of musk [8]. Here, we speculate that one of the reasons why muskrat musk is closer to WM may be due to the fact that muskrat musk was not fermented in the pod for enough time when we collected muskrat musk by abdominal extrusion. In addition, the milk white appearance of muskrat musk is also close to that of WM, which may be caused by the similar composition. Further studies are needed to investigate the relationship between the composition changes of muskrat musk and microorganisms. The top 20 metabolites in each sample were selected according to their relative content (Table 1), and 6 metabolites were found to be co-expressed (Figure 2c). They are classified into amino acids (alanine, 4-oxoproline, and glycine) and long-chain fatty acids (palmitic acid, stearic acid, and oleic acid). Li et al. found that there was a tendency for amino acids to be enriched in the sachets during the muskrat’s secretory phase [14], a finding similar to the results of Zhang et al. on musk [7]. Based on amino acid enrichment in amino acid biosynthesis and amino acid metabolism pathways (Figure 5e), our hypothesis is consistent that the enrichment of amino acids in the preputial gland may provide support for their musk secretion and reproductive behavior. In addition, we speculate that amino acids in the preputial gland are also involved in protein and peptide synthesis based on the aminoacyl t-RNA biosynthesis pathway (Figure 5e). However, the exact mechanism needs to be further studied. Long-chain fatty acids, such as palmitic acid and stearic acid, are the main fatty acids used for energy storage in animal tissues and generally produce energy through the β-oxidation-mediated degradation of acetyl-CoA [27]. In addition, fatty acids are converted to ketone bodies through a series of enzymatic reactions in the mammalian liver [28]. Previous studies have shown that human and mouse retinal epithelial cells metabolize fatty acids to produce ketones, which are released into the retinal microenvironment [29]. Fatty acids are important components of muskrat musk and musk. Understanding whether they are produced by muskrat epithelial cells as well as their conversion to ketone bodies will be the focus of future studies.

Next, we conducted enrichment and topological analyses of the SDMs to screen the key pathways with the highest correlation to the metabolite differences. In our study, the increase in metabolites with age was mainly concentrated in amino acid biosynthesis and metabolism, steroid hormone biosynthesis, and the fatty acid biosynthesis pathway, which is consistent with the results of Li et al. [14]. Valine, leucine, and isoleucine are branched-chain amino acids (BCAAs). They can more efficiently generate ATP than other amino acids and act as substrates for protein synthesis [30]. Alanine, aspartic acid, and glutamate metabolism play an important role in the conversion of glutamic acid to pyruvate. Pyruvate also plays a central role in the interconversion of sugars, fats, and amino acids through the acetyl-CoA and citric acid cycles [31]. In addition, studies have shown that glycine, serine, and threonine metabolism can provide raw materials for the metabolism of the aforementioned valine, leucine, and isoleucine biosynthesis as well as alanine, aspartate, and glutamate metabolism [32]. Therefore, we speculated that the enrichment of amino acid metabolism during the secretion stage may ensure the energy supply for musk secretion. However, changes in the metabolic pathways of the other remaining amino acids need further study. Interestingly, steroid hormone biosynthesis is a unique pathway found in the samples. Generally, the optimal breeding age of muskrats is 1 year old, with most muskrats living 4 to 5 years. With the physical decline caused by age, the levels of sex hormones also drop [33]. Although testosterone levels decrease with age, other hormones, such as pregnenolone and dihydrotestosterone, show an upward trend. Future research should determine the factors that regulate the high expression of these sex hormones in middle and later life.

## 5. Conclusions

In summary, the current study used GC–TOF–MS to analyze the variation trend of muskrat musk composition and its matching degree with musk at different ages. These analyses confirmed that muskrat musk has higher similarity with WM, but there is still a gap compared with BM. The identified differential metabolites and pathways were closely related to the development of muskrat glands and the secretion of muskrat musk, which confirmed that energy and hormones play a leading role in the breeding season. These data can inform the search for alternatives to musk, determine the best time to collect muskrat musk, and support the rapid development of muskrat farming.

## Figures and Tables

**Figure 1 animals-13-01678-f001:**
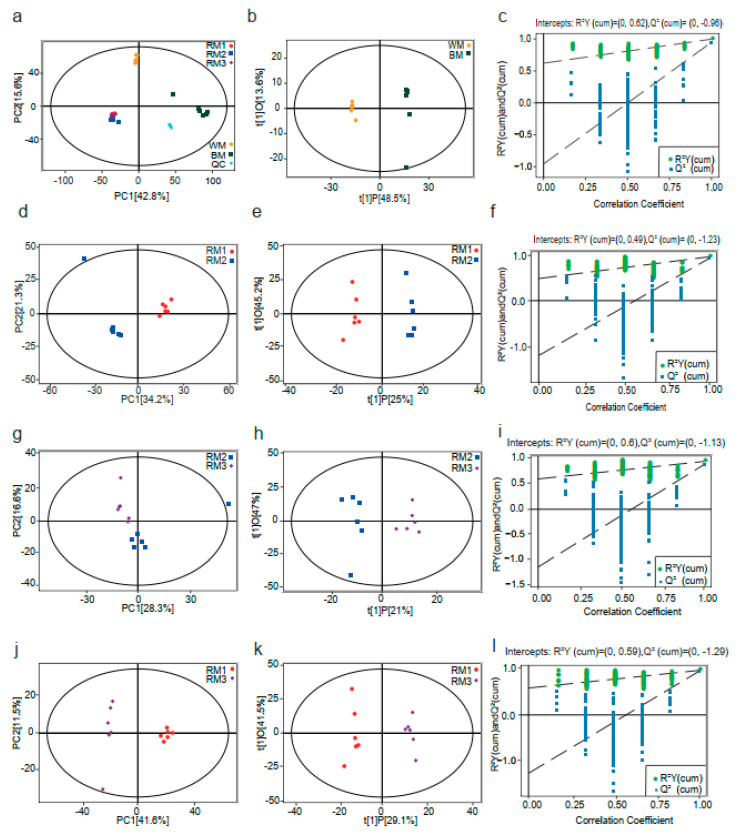
PCA model score scatter plots, OPLS–DA model, and permutation test. (**a**,**d**,**g**,**j**) The PCA models; (**b**,**e**,**h**,**k**) OPLS–DA models; and (**c**,**f**,**i**,**l**) permutation tests of the OPLS–DA models were derived from the GC–MS metabolomics profiles of musk from muskrats of different ages. Note: RM1: muskrat musk from 1-year-old muskrat, RM2: muskrat musk from 2-year-old muskrat, RM3: muskrat musk from 3-year-old muskrat, WM: white musk from forestry musk deer, and BM: brown musk from forestry musk deer.

**Figure 2 animals-13-01678-f002:**
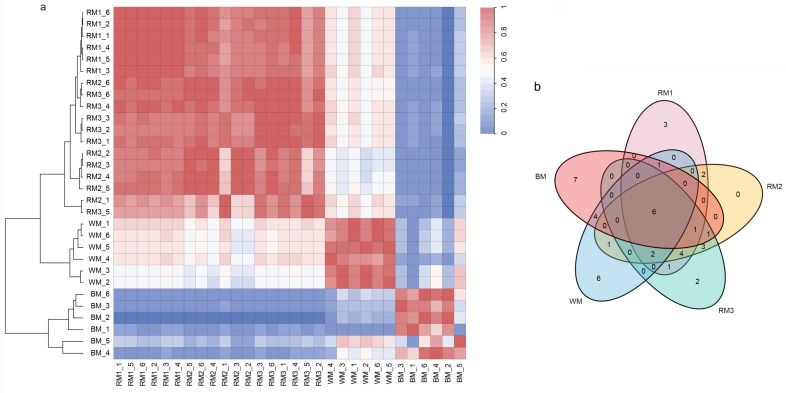
Correlation analysis between muskrat musk and musk. (**a**) Show correlations and clustering between different samples and between different replicates of the same sample (correlation calculation method = Pearson’s r, clustering method = ward. D2); (**b**) Comparative analysis of the top 20 metabolites among 5 samples. Note: RM1_1-RM1_6 = 6 repeats from 1-year-old muskrat, RM2_1-RM2_6 = 6 repeats from 2-year-old muskrat, RM3_1-RM3_6 = 6 repeats from 3-year-old muskrat, WM_1-WM_6 = 6 repeats of white musk, and BM_1-BM_6 = 6 repeats of brown musk.

**Figure 3 animals-13-01678-f003:**
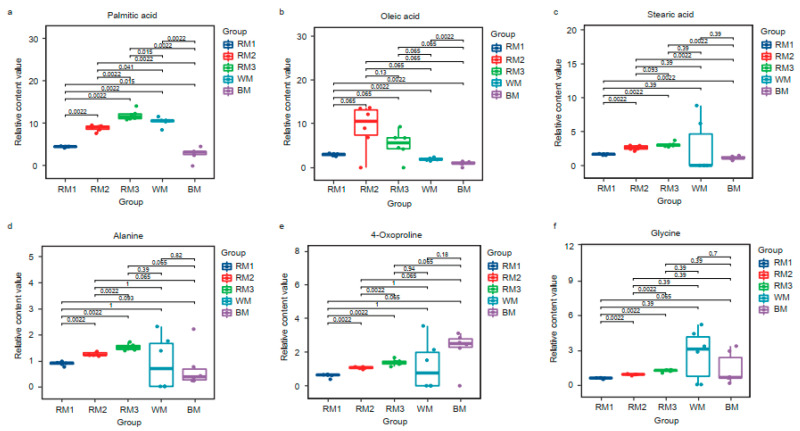
Box plot of the same components in different types of samples. (**a**) Palmitic acid; (**b**) Oleic acid; (**c**) Stearic acid; (**d**) Alanine; (**e**) 4-Oxoproline; (**f**) Glycine. Note: Each group had 6 replicates. The larger the P value between groups, the better the groups’ matching degree. The statistical method was a t-test. RM1: muskrat musk from 1-year-old muskrat, RM2: muskrat musk from 2-year-old muskrat, RM3: muskrat musk from 3-year-old muskrat, WM: white musk from forestry musk deer, BM: brown musk from forestry musk deer.

**Figure 4 animals-13-01678-f004:**
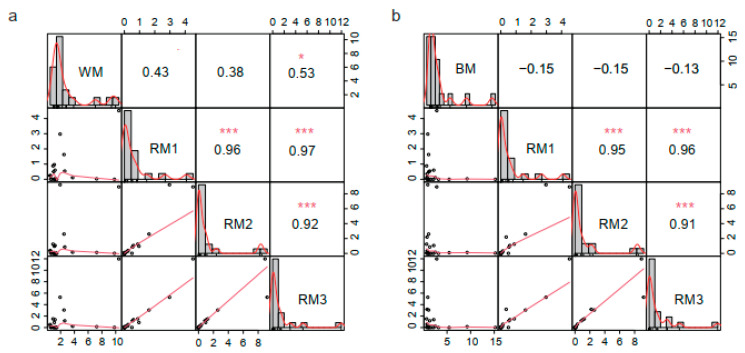
Correlation analysis of the top 20 metabolites between groups. (**a**) Correlation analysis between WM and RM1-RM3; (**b**) Correlation analysis between BM and RM1-RM3. Note: Correlation coefficient method = Pearson’s r, * *p* < 0.05, *** *p* < 0.001. RM1: muskrat musk from 1-year-old muskrat, RM2: muskrat musk from 2-year-old muskrat, RM3: muskrat musk from 3-year-old muskrat, WM: white musk from forestry musk deer, BM: brown musk from forestry musk deer.

**Figure 5 animals-13-01678-f005:**
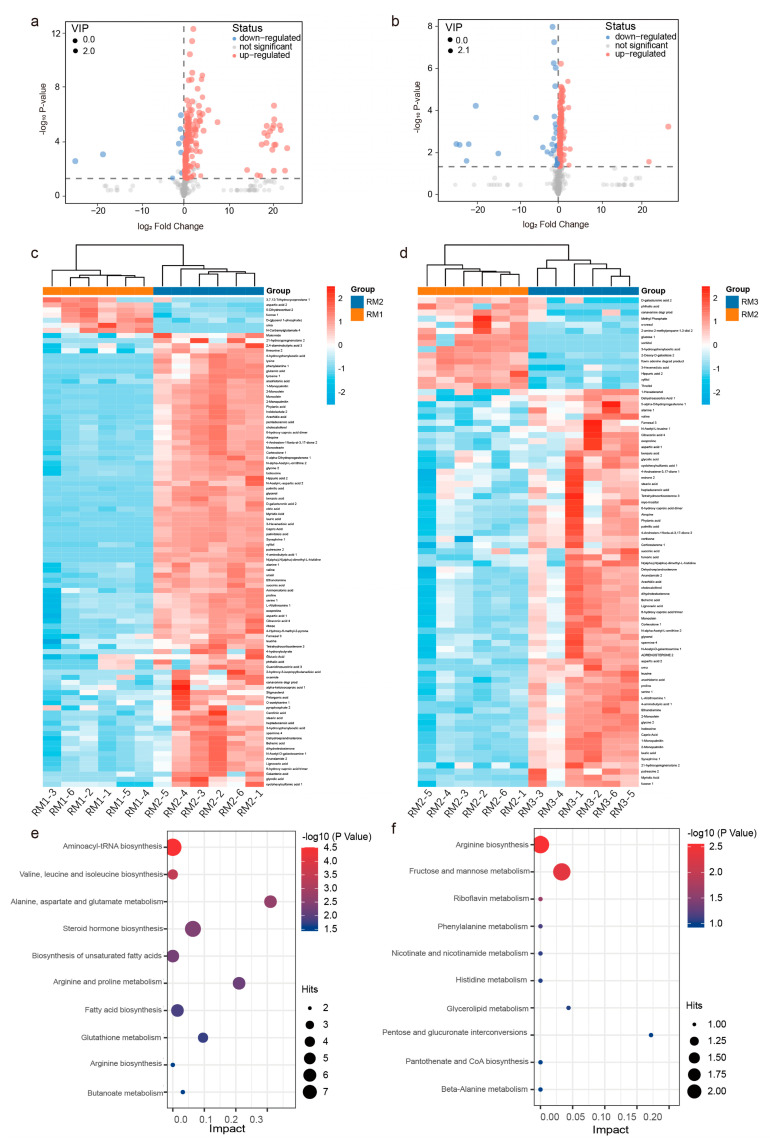
Functional enrichment analysis of significantly different metabolites. (**a**) Volcano plots of the RM1 vs. RM2 and (**b**) RM2 vs. RM3. (**c**) Heat map and cluster analysis of the RM1 vs. RM2 and (**d**) RM2 vs. RM3. Top 10 pathways of metabolite enrichment: (**e**) up-regulated and (**f**) down-regulated. Note: RM1_1-RM1_6 = 6 repeats from 1-year-old muskrat, RM2_1-RM2_6 = 6 repeats from 1-year-old muskrat, and RM3_1-RM3_6 = 6 repeats from 1-year-old muskrat.

**Table 1 animals-13-01678-t001:** Chemical constituents of muskrat musk and Chinese forest musk deer preputial gland secretion from (methanol/chloroform (*v*:*v*) = 3:1) extraction.

No.	Chemical Name	CAS	Molecular Formula	RM1	RM2	RM3	WM	BM
1	Palmitic acid	57-10-3	C_16_H_32_O_2_	4.5182	8.9303	11.9766	10.4605	2.8420
2	Oleic acid	112-80-1	C_18_H_34_O_2_	2.9752	9.2118	5.2908	1.9322	0.9327
3	Stearic acid	57-11-4	C_18_H_36_O_2_	1.6265	2.5967	3.0584	2.5001	1.1093
4	2-Hydroxypyridine	142-08-5	C_5_H_5_NO	0.9613	0.9644	0.8798	1.0861	0.6629
5	Alanine	56-41-7	C_3_H_7_NO_2_	0.8807	1.2418	1.5152	0.9043	0.6899
6	Testosterone	58-22-0	C_19_H_28_O_2_	0.7483	0.2393	7.6711 × 10^−9^	9.8221 × 10^−9^	1.1148 × 10^−8^
7	4-Oxoproline	2002-02-0	C_5_H_7_NO_3_	0.6002	1.0628	1.3813	1.1985	2.1974
8	Glycine	56-40-6	C_2_H_5_NO_2_	0.5406	0.8617	1.1840	2.6123	1.3521
9	D-Arabitol	488-82-4	C_5_H_12_O_5_	0.5005	0.0125	0.0229	1.0005	1.1348
10	Leucine	61-90-5	C_6_H_13_NO_2_	0.4116	0.5335	0.7207	0.1310	0.0668
11	Glycerol	56-81-5	C_3_H_8_O_3_	0.3551	2.1961	3.1816	0.4295	0.7871
12	D-Fucose	2438-80-4	C_6_H_12_O_5_	0.3512	0.2863	0.3478	0.2382	0.0343
13	Glucose	50-99-7	C_6_H_12_O_6_	0.3226	0.2003	7.6712 × 10^−9^	9.8221 × 10^−9^	1.1148 × 10^−8^
14	Valine	72-18-4	C_5_H_11_NO_2_	0.3089	0.5364	0.6613	0.4140	0.1031
15	Canavanine	543-38-4	C_5_H_12_N_4_O_3_	0.2698	0.3700	0.1309	0.2788	0.4252
16	Serine	56-45-1	C_3_H_7_NO_3_	0.2481	0.3993	0.5218	0.4868	0.1616
17	4-Androstene-3,17-dione	63-05-8	C_19_H_26_O_2_	0.2438	0.2540	0.2944	9.8221 × 10^−9^	1.1148 × 10^−8^
18	Palmitoleic acid	373-49-9	C_16_H_30_O_2_	0.2320	2.9838	4.4162	0.0930	0.0528
19	Isoleucine	73-32-5	C_6_H_13_NO_2_	0.2223	0.3772	0.5495	0.0727	0.0530
20	Urea	57-13-6	CH_4_N_2_O	0.2188	7.9282 × 10^−9^	0.8148	0.0423	0.0386
21	2,4-Diaminobutyric acid	305-62-4	C_4_H_10_N_2_O_2_	0.2176	0.2830	0.2657	0.5681	1.1148 × 10^−8^
22	Proline	147-85-3	C_5_H_9_NO_2_	0.1821	0.2990	0.4460	0.3287	1.1148 × 10^−8^
23	Atropine	51-55-8	C_17_H_23_NO_3_	0.1617	0.2891	0.4272	9.8221 × 10^−9^	0.0071
24	6-Hydroxy caproic acid trimer	1191-25-9	C_6_H_12_O_3_	0.1520	0.2782	0.4126	0.3575	0.1047
25	Arachidic acid	506-30-9	C_20_H_40_O_2_	0.1407	0.3384	0.5872	0.2763	0.7918
26	Behenic acid	112-85-6	C_22_H_44_O_2_	0.1088	0.1580	0.2217	3.7522	2.5273
27	Monoolein	111-03-5	C_21_H_40_O_4_	0.0475	0.4066	0.6111	0.0028	0.0362
28	Lignoceric acid	557-59-5	C_24_H_48_O_2_	0.0391	0.0642	0.0950	1.1968	0.7426
29	Succinic acid	110-15-6	C_4_H_6_O_4_	0.0291	0.1264	0.1512	7.2419	9.2730
30	1-Monopalmitin	542-44-9	C_19_H_38_O_4_	0.0123	0.2774	0.6080	0.0465	1.1148 × 10^−8^
31	4-Hydroxyphenylacetic acid	156-38-7	C_8_H_8_O_3_	0.0084	0.0599	0.0637	0.5719	1.0516
32	Carbobenzyloxy-L-leucine degr	2018-66-8	C_14_H_19_NO_4_	8.1136 × 10^−9^	0.0425	0.0216	0.0434	15.2327
33	Hippuric acid	495-69-2	C_9_H_9_NO_3_	8.1136 × 10^−9^	0.0989	0.0024	9.8221 × 10^−9^	5.4626
34	Threo-beta-hyrdoxyaspartate	7298-99-9	C_4_H_7_NO_5_	8.1136 × 10^−9^	0.0009	7.6712 × 10^−9^	0.0021	3.2056
35	Hypoxanthine	68-94-0	C_5_H_4_N_4_O	8.1136 × 10^−9^	7.9282 × 10^−9^	0.0019	9.8221 × 10^−9^	2.1271
36	Methyl trans-cinnamate	103-26-4	C_10_H_10_O_2_	8.1136 × 10^−9^	7.9282 × 10^−9^	0.0005	9.8221 × 10^−9^	1.6237
37	Biuret	108-19-0	C_2_H_5_N_3_O_2_	8.1136 × 10^−9^	7.9282 × 10^−9^	7.6712 × 10^−9^	9.8221 × 10^−9^	1.2075
38	1-Octanal	124-13-0	C_8_H_16_O	8.1136 × 10^−9^	7.9282 × 10^−9^	7.6712 × 10^−9^	9.8221 × 10^−9^	0.7773
39	Thymidine	50-89-5	C_10_H_14_N_2_O_5_	8.1136 × 10^−9^	7.9282 × 10^−9^	7.6712 × 10^−9^	1.3371	0.5811
40	1-Hexadecanol	36653-82-4	C_16_H_34_O	8.1136 × 10^−9^	0.0127	0.0329	9.8164	0.3257
41	5-Aminovaleric acid	660-88-8	C_5_H_11_NO_2_	8.1136 × 10^−9^	7.9282 × 10^−9^	7.6712 × 10^−9^	1.4447	1.1148 × 10^−8^
42	Putrescine	110-60-1	C_4_H_12_N_2_	8.1136 × 10^−9^	0.0039	0.0087	1.2105	1.1148 × 10^−8^
43	Octadecanol	112-92-5	C_18_H_38_O	8.1136 × 10^−9^	7.9282 × 10^−9^	7.6712 × 10^−9^	1.0077	1.1148 × 10^−8^
44	Glucoheptonic acid	23351-51-1	C_7_H_14_O_8_	8.1136 × 10^−9^	7.9282 × 10^−9^	7.6712 × 10^−9^	0.6318	1.1148 × 10^−8^

Note: The table only lists the top 20 extracts’ relative content of each group. The relative content (%) is the mean value of six replicates per group. The “gray marks” in the table represent the top 20 extracts from each group.

## Data Availability

All data generated or analyzed during this study are included.

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
