# Peer review of "Correlation Analysis between Muskrat (Ondatra zibethicus) Musk and Traditional Musk"

_animals, 2023, doi:10.3390/ani13101678_

Round 1
Reviewer 1 Report
General comments:
Musk is a traditional Chinese medicine ingredient of considerable economic value. Yet, it is derived from the endangered species of forest musk deer, rendering it of paramount importance to identify new sources of musk. Thus, the aim of this study was to examine the potential of muskrat musk to serve as a substitute for musk by comparing their components. This research provides a vital foundation for the protection of the endangered species of forest musk deer. Nevertheless, the introduction should include information pertaining to muskrats, such as their protection grade, degree of endangerment, and the current situation of domestic breeding. We cannot destroy muskrats in the name of conserving musk deer. In the discussion, the authors ought to discuss how to resolve this dilemma. If muskrat musk were to be utilized in place of musk, it should be noted that muscone is a crucial active ingredient of musk, but muskrat musk does not contain muskone. The author ought to elucidate this enigma in the article as well.
Special comments:
Line 164-165: The authors indicated a greater correlation between BM and WM than muskrat musk. Nevertheless, as demonstrated by the heat map (Fig.2a), when compared with BM, WM evidently has a higher correlation with muskrat incense. Please confirm it.
Line 181: The metabolites of musk top20 should be listed and can be appended as a supplementary file.
Line 217: Figure 4f —> Figure 5f
Author Response
We appreciate your perusal of our paper and your insightful comments. The following adjustments have been implemented as a result of your comments.
General comments:
Point 1: Nevertheless, the introduction should include information pertaining to muskrats, such as their protection grade, degree of endangerment, and the current situation of domestic breeding.
Response 1: After consulting the relevant literature, we added the protection level of muskrat and the current situation of domestic breeding in the introduction (Line 58-61).
Point 2: We cannot destroy muskrats in the name of conserving musk deer. In the discussion, the authors ought to discuss how to resolve this dilemma.
Response 2: The reviewer said “We cannot destroy muskrats in the name of conserving musk deer.” This is a very forward-looking idea. Thus, we appeal in the discussion that a conservation system for muskrat should be established, as well as a farming system that considers animal welfare (Line 282-285).
Point 3: If muskrat musk were to be utilized in place of musk, it should be noted that muscone is a crucial active ingredient of musk, but muskrat musk does not contain muskone. The author ought to elucidate this enigma in the article as well.
Response 3: Thanks for giving us valuable advice. We did not detect muscone in muskrat musk in this experiment, including our unpublished study (GC analysis of muskrat musk from five regions, six samples from each region were repeated, and only trace amounts of muscone were detected in one sample from one of the regions, indicating that muscone content in muskrat musk is relatively low). However, different active substances may also exert the same pharmacological function, which we supplemented it in the discussion (Line 278-281).
Special comments:
Point 4: Line 164-165: The authors indicated a greater correlation between BM and WM than muskrat musk. Nevertheless, as demonstrated by the heat map (Fig.2a), when compared with BM, WM evidently has a higher correlation with muskrat incense. Please confirm it.
Response 4: Sorry, our previous description was ambiguous and has been revised to “When opposed to RM, BM had a greater correlation with WM”.
Point 5: Line 181: The metabolites of musk top20 should be listed and can be appended as a supplementary file.
Response 5: Musk top20 metabolites were listed as supplementary tables (Line 469-471).
Point 6: Line 217: Figure 4f —> Figure 5f
Response 6: Figure 4f —> Figure 5f
Reviewer 2 Report
Shi et al. reported the correlation analysis between muskrat (Ondatra zibethicus) musk and traditional musk. Musk is widely used in traditional Chinese medicine. This paper indicated that muskrat musk from three-year-old 32 muskrats is a relatively good substitute for white musk, and the result also implies to amino acid 33 biosynthesis and metabolism, steroid hormone biosynthesis, and fatty acid biosynthesis is beneficial 34 to the secretion of muskrat musk. In general, the topic is interesting, and the results are reasonable. However, there are some issues in this manuscript. The first time the abbreviation appears, the author should provide the full name, such as GC-TOF-MS, and RSD in the text. Detailed information on male forest musk deer should be provided in the Materials and Methods. Figure 3 and Figure 4 contain sub-figures. The information for each sub-figure should be provided.
The topic is original and addresses a specific gap in the field.
The conclusions are consistent with the evidence and arguments presented
and they address the main question posed.
Author Response
We thank the reviewer very much for his comment and affirmation of our study.
General comments:
Shi et al. reported the correlation analysis between muskrat (Ondatra zibethicus) musk and traditional musk. Musk is widely used in traditional Chinese medicine. This paper indicated that muskrat musk from three-year-old 32 muskrats is a relatively good substitute for white musk, and the result also implies to amino acid 33 biosynthesis and metabolism, steroid hormone biosynthesis, and fatty acid biosynthesis is beneficial 34 to the secretion of muskrat musk. In general, the topic is interesting, and the results are reasonable. However, there are some issues in this manuscript. The first time the abbreviation appears, the author should provide the full name, such as GC-TOF-MS, and RSD in the text. Detailed information on male forest musk deer should be provided in the Materials and Methods. Figure 3 and Figure 4 contain sub-figures. The information for each sub-figure should be provided.
For the questions raised by the reviewer, we will answer one-to-one as follows:
Point 1: The first time the abbreviation appears, the author should provide the full name, such as GC-TOF-MS, and RSD in the text.
Response 1: We thank the reviewer’s careful review of the manuscript. We have revised the manuscript. Full name of the supplementary “GC-TOF-MS” —> “Gas Chromatography Time-of-Flight Mass Spectrometry” (Line 17), “RSD” —> “Relative Standard Deviation” (Line 139).
Point 2: Detailed information on male forest musk deer should be provided in the Materials and Methods.
Response 2: Similar questions were raised by other reviewers, and this is indeed important information that we missed. Line 86-90: Here, we provide the composition of the daily diet of muskrat and forest musk deer.
Point 3: Figure 3 and Figure 4 contain sub-figures. The information for each sub-figure should be provided.
Response 3: Line 229-230, 236-237: Here, the legends of Figure 3 and Figure 4 subfigures have been supplemented.
Reviewer 3 Report
Dear Authors,
Your study is quite interesting, especially to someone keenly interested in animals and animal-related products.
To make it more readable, you need to carefully check the English wording throughout your paper. For example, in line 25, you should say were picked instead of were pick.
My comments are about the content and construct of this paper; I did not conduct a review of the literature or compare it with other published material about this area.
The conclusions are consistent with the evidence presented
I have no comments on figures and tables.
Best, A Reviewer
Author Response
We sincerely thank the reviewer’s suggestions on our manuscript.
Special comments:
Point 1: To make it more readable, you need to carefully check the English wording throughout your paper. For example, in line 25, you should say were picked instead of were pick.
Response 1: We have verified and altered our English wording once again in accordance with your advice.
Line 26: “were pick” —> “were picked”
Line 29: “7” —> “Seven”
Line 139: “an” —> “a”
Reviewer 4 Report
The paper presents an innovative approach to the research of metabolomic changes of muskrat during different ages and It is suitable for publication in a scientific journal.
The main complaint is related to the relatively small number of samples. The research started with 10 individuals, and in the end only six individuals were sampled. Such a small sample does not necessarily mean that the research is weak, however, all presented results must be taken conditionally. It would be desirable to check the manuscript by one of the statistical editors. It would be good if the experiment was designed with groups of differently fed animals, classified by sex and age category. Here we are dealing with six animals that are periodically sampled. There is also no control group of animals that were fed/kept in somewhat different conditions. This design contributed to the reliability of the displayed data
In chapter 2.1, it would be interesting to describe what food the animals were fed in a certain season. Food can have an influence on the researched parameters (fatty acids, their relationship, etc.), so I think it is necessary to describe the nutrition of animals in more detail. Also, although there were only six individuals, the gender/sex ratio should be highlighted. Did the authors find a difference in the investigated parameters between the sexes?
in chapter 3.3, the authors talk about correlational comparisons of two species of muskrat. How was the data that was compared obtained, what is the source of the data used in the comparison?
Some of the graphic representations are unreadable (4c; 4d)
Author Response
We sincerely thank the reviewer’s valuable suggestions on our manuscript, and the following is our answer to your suggestions and questions.
Point 1: It would be good if the experiment was designed with groups of differently fed animals, classified by sex and age category. Here we are dealing with six animals that are periodically sampled. There is also no control group of animals that were fed/kept in somewhat different conditions. This design contributed to the reliability of the displayed data.
Response 1: Because there is no finished forest musk deer and muskrat feed sold in the market and there are no specific nutritional requirements for forest musk deer and muskrat at each stage, the feed was not classified according to different seasons and different stages in this study.
Point 2: In chapter 2.1, it would be interesting to describe what food the animals were fed in a certain season. Food can have an influence on the researched parameters (fatty acids, their relationship, etc.), so I think it is necessary to describe the nutrition of animals in more detail.
Response 2: That's really helpful advice. Thank you very much. Food does affect the composition and content of metabolites, but our nearly identical dietary formulations for muskrat and forest musk deer allowed for their comparative analysis. Line 86-90: Here, we provide the composition of the daily diet of muskrat and forest musk deer.
Point 3: Also, although there were only six individuals, the gender/sex ratio should be highlighted. Did the authors find a difference in the investigated parameters between the sexes?
Response 3: Thanks for your advice. Only male muskrats have muskrat glands that can secrete muskrat incense, which we supplement in Subsection 2.1 for general readers to understand (Line 83, 93).
Point 4: in chapter 3.3, the authors talk about correlational comparisons of two species of muskrat. How was the data that was compared obtained, what is the source of the data used in the comparison?
Response 4: The data for the correlation analysis between muskrat musk and musk in Chapter 3.3 were derived from the relative content of co-expressed metabolites obtained from GC-TOF-MS metabolomics analysis (Table S1). The top metabolites were ranked according to the magnitude of their relative content.
Point 5: Some of the graphic representations are unreadable (4c; 4d)
Response 5: After checking, there is no 4c or 4d in our Figure 4.